# Influence of Various Factors on Caffeine Content in Coffee Brews

**DOI:** 10.3390/foods10061208

**Published:** 2021-05-27

**Authors:** Ewa Olechno, Anna Puścion-Jakubik, Małgorzata Elżbieta Zujko, Katarzyna Socha

**Affiliations:** 1Department of Food Biotechnology, Faculty of Health Science, Medical University of Białystok, Szpitalna 37 Street, 15-295 Białystok, Poland; ewaolechno1996@gmail.com (E.O.); malgorzata.zujko@umb.edu.pl (M.E.Z.); 2Department of Bromatology, Faculty of Pharmacy with the Division of Laboratory Medicine, Medical University of Białystok, Mickiewicza 2D Street, 15-222 Białystok, Poland; katarzyna.socha@umb.edu.pl

**Keywords:** caffeine, Arabica, Robusta, water, pressure, coffee roasting, origin, temperature of water, coffee/water ratio, brewing time

## Abstract

Coffee brews are one of the most popular drinks. They are consumed for caffeine and its stimulant properties. The study aimed to summarize data on the influence of various factors on caffeine content in brews prepared with different methods. The study was carried out using a literature review from 2010–2020. PubMed and Google Scholar databases were searched. Data on caffeine content was collected by analyzing the following factors: the influence of species, brewing time, water temperature, pressure, degree of roast, grinding degree, water type, water/coffee ratio as well as other factors (such as geographical origin). To sum up, converting caffeine content to 1 L of the brew, the highest content is that of brews prepared in an espresso machine (portafilter), with the amount of 7.5 g of a coffee blend (95% Robusta + 5% Arabica), and water (the volume of coffee brew was 25 mL) at a temperature of 92 °C and a pressure of 7 bar, but the highest content in one portion was detected in a brew of 50 g of Robusta coffee poured with 500 mL of cold water (25 °C) and boiled.

## 1. Introduction

The history of coffee began in Ethiopia, former Abyssinia [1] and this beverage is consumed by communities around the world [2]. Coffee belongs to the Rubiaceae family, the fourth largest angiosperm family, consisting of 124 species spread over two genera, Coffea and Psilanthus [3]. Among them, Arabica—*Coffea arabica* L. and Robusta coffee—*Coffea canephora* Pierre ex Froehner, are of major commercial importance. Most commercial coffee sold as ‘robusta’ is not *Coffea canephora* var. *robusta*, but of other varieties (probably mostly hybrids) [3,4,5]. Both the production and consumption of coffee are constantly increasing. According to data from 2020/2021, it has increased by 1.1% compared with 2017/2018 [6]. In 2019, coffee production reached about 9,903,180 tons, and in 2020 there was an increase by 6.4%—about 10,538,820 tons. In addition, the production of Arabica is greater than that of Robusta. According to data from 2020, production of Arabica coffee was 6,319,500 tons (an increase by 13.6% compared with the previous year), and of Robusta—4,219,380 tons (a decrease by 2.8%) [7].

Coffee consists of over 1.000 bioactive substances [8]. On a dry weight basis, Arabica and Robusta green beans contain, respectively: polysaccharides (50–55% and 37–47%), oligosaccharides (6–8% and 5–7%), lipids (12–18% and 9–13%), proteins (11–13%), chlorogenic acids (5.5–8% and 7–10%), minerals (3–4.2% and 4–4.5%), fatty acids (1.5–2%), caffeine (0.9–1.2% and 1.6–2.4%), trigonelline (1–1.2% and 0.6–0.8%) and free amino acids (2%). The composition of roasted coffee beans differs from the above, for Arabica: polysaccharides (24–39%), oligosaccharides (0–3.5%), proteins (13–15%), chlorogenic acids (1.2–2.3%), free amino acids (0%), lipids (14.5–20%), minerals (3.5–4.5%), fatty acids (1–1.5%), trigonelline (0.5–1%), caffeine (0–1%), and melanoidins (16–17%) formed in the process of roasting coffee beans [9]. 

A review of the literature confirms the beneficial effects of coffee on health. Coffee consumption has been shown to correlate, among other things, with lower incidence of: neurodegenerative diseases, death from cerebro- and cardiovascular causes, cancer, especially endometrial cancer, prostate cancer, leukemia, melanoma, and non-melanoma skin cancer, oral cancer, and liver cancer, and other liver diseases such as non-alcoholic fatty liver disease, liver fibrosis, and cirrhosis, but also type 2 diabetes and metabolic syndrome [10,11,12]. Coffee is not recommended to people who suffer from stomach diseases, such as gastroesophageal reflux, peptic ulcer disease, or acute gastritis [13,14,15,16]. It should also be limited to pregnant and lactating women due to the lack of sufficient research in this area [17,18,19,20,21]. 

The reasons for consuming coffee include the desire to improve cognitive abilities and concentration, reduce fatigue and sleepiness. These properties are determined by the presence of caffeine in coffee [22]. 

Caffeine is an alkaloid, a secondary plan metabolite, that is an antagonist of adenosine receptors: A1 and A2. This has a stimulating effect on the centers of the nervous system [23,24]. Arabica green beans contain, on average, 0.9 to 1.5% dry weight of caffeine. In contrast, Robusta green beans have between 1.2 and 2.4% of the alkaloid [9,25,26,27,28]. In plants, this substance acts as a protection against insects [29]. 

Caffeine is demethylated in the liver. The following metabolites are then produced: paraxanthin, theobromine, and theophylline [30,31]. Half-life time of caffeine in plasma is from 2.5 to 5.0 h [32] and depends on age, gender, use of certain medications such as oral contraceptives (which increase its by 5–10 h), carbamazepine, rifampicin (shortens), cimetidine, or ciprofloxacin (increases) and physiological states e.g., pregnancy, smoking, and liver diseases are associated to the increase of half-life time of caffeine [30,33,34]. Caffeine’s metabolism is genetically determined. It is metabolized by special enzymes known as 1A2 or CYP1A2 [31,35]. This means that the speed with which the substance is removed varies from one individual to another, as it depends on the presence of one of the alleles—CYP1A2 * 1F or CYP1A2 * 1A. Some people who metabolize caffeine slowly may experience nausea, weakness, palpitations, or anxiety after consumption, which will not be experienced by people with rapid metabolism of this substance [30,36]. Caffeine metabolism may also be related to the frequency and manner of coffee consumption. People with a lower metabolic rate (measuring caffeine from saliva) have been shown to consume less coffee a day and add sugar more frequently [37]. The content of individual ingredients, including caffeine, also depends on the type and origin of coffee, the place of cultivation and the type of soil associated with it, the method of cultivation, climatic and environmental conditions, the processing of the beans, i.e., the cleaning and roasting process, as well as the time and conditions of storage [36,38,39,40,41,42,43,44,45,46]. The differences reported in the caffeine content may also result from the method of its measurement [47].

According to the European Food Safety Authority (EFSA), the daily consumption of caffeine by a healthy adult should not exceed 400 mg during the day, while a single dose of caffeine should not exceed 3 mg/kg body weight. Pregnant and breastfeeding women should not consume more than 200 mg per day [48,49].

Moreover, the content of bioactive substances in coffee beans can differ from the amount that will remain in the brew. Additional factors that affect the content of substances in brews are the method of brewing, including the grinding thickness, extraction time, the amount of water, the temperature of water, vapor pressure in the case of espresso coffee, and coffee/water ratio [26,50,51,52,53,54,55,56,57,58,59,60].

There are many brewing methods more or less different from one another. Some of them are: pouring ground coffee with hot or cold water, brewing in a coffee machine, filter coffee machine, portafilter, French press, Aeropress, Neapolitan pot or dripper [52,61,62,63,64,65,66,67,68,69,70,71,72,73]. Depending on the brewing technique, the consumer can drink coffee with a completely different taste, aroma, and biochemical composition [39,53,58,60,74,75,76,77,78].

This study aimed to review the literature with regard to the assessment of factors influencing caffeine content in the coffee brew.

The study takes into account research from 2010–2020. The databases searched were: Google Scholar and PubMed. The following terms were searched: ‘coffee’, ‘Arabica’, ‘Robusta’, ‘caffeine, ‘coffee beans’, ‘coffee brewing methods’, ‘coffee origin’, ‘time of brewing coffee’, ‘espresso’, ‘type of water and caffeine’, ’roasting process’, and ‘degree of grinding’. The inclusion criteria included: species and type of coffee, the origin of coffee, description of the brewing methods, and degree of roasting, while exclusion criteria included: lack of type or species of coffee and preparation methods that deviates from domestic conditions.

## 2. Factors Affecting Caffeine Content in Different Coffee Beverages

Table 1, Table 2 and Table 3 present the literature data obtained during the research review. Table 1 contains information about Arabica coffee brews, Table 2—Robusta coffee brews, and Table 3—blends of Arabica and Robusta coffee brews. The coffees marked as ‘Robusta’ may be varieties that are not *Coffea canephora* var. robusta, but the tables were made based on the data provided by the authors of the publications being the subject of this review. The results in the tables are ranked from the most recent publications.

To compare results by other authors, the presentation of caffeine content in brews was converted for g/L (caffeine content in the finished portion). Among the Arabica varieties, the highest caffeine content (7.908 g/L) was detected by Ludwig et al. (2014) [70] in an espresso; the lowest—0.006 g/L (decaffeinated coffee)—was recorded by Macheiner (2019) [63], in a brew of Arabica in hot water. In the case of Robusta, the highest caffeine content (2.581 g/L) was found in espresso coffee in the study by Fărcaş et al. (2014) [69], while the lowest content (0.150 ± 0.010 g/L) was found in espresso [65]. For Arabica and Robusta blends, Caprioli et al. (2015) [67] obtained the highest content of the substance in question in espresso coffee: 10.303 g/L. 

Taking into account the amount of roasted coffee used, an espresso of Arabica coffee in the study by Ludwig (2014) [70] had the highest concentration obtained with the least amount of coffee (4.218 g/L). 

Green coffee poured with hot water obtained the following values: from 0.006 to 0.188 ± 0.007 g/L of brew (Arabica) and from 0.186 ± 0.008 to 0.293 ± 0.014 g/L of brew (Robusta) [63].

### 2.1. The Impact of Species

The most produced and consumed coffees, known as Arabica and Robusta, differ significantly in their caffeine content. Robusta (including all varieties of *Coffea canephora*) contains more caffeine than Arabica. It is a less valued variety on the world market [26,79]. These two varieties also differ in terms of their cultivation and resistance to diseases and pests [80].

Caffeine is formed in unripe coffee beans and its amount increases as they mature [81]. The higher content of caffeine in Robusta coffee is due to the greater expression of certain genes, such as CaXMT1, CaMXMT1, and CaDXMT2, which are associated with caffeine accumulation in coffee beans [38,82].

In the study by Fărcaş et al. (2014) [69], a brew of Robusta coffee prepared in the coffee machine was characterized by a higher content of caffeine: 2.581 g/L (0.258/100 mL of Robusta brew), while the content of caffeine in a brew of Arabica coffee was 1.876 g/L. The variables used were the same. The Robusta brew contained almost 1.4 times more caffeine than the Arabica brew. 

In the study by Merecz et al. (2018) [65], a brew of Arabica prepared in a coffee machine contained surprisingly more caffeine than a brew of Robusta: from 0.330 ± 0.020 to 0.410 ± 0.020 g/L (Arabica, various sources of origin) and 0.150 ± 0.010 g/L (Robusta). In contrast, in the same study, Robusta contained more caffeine than samples of Arabica brews made by pouring hot water and in a percolator. Other literature reports indicate that Arabica has less caffeine, therefore the question requires further research.

In a study by Macheiner et al. (2019) [63], where caffeine content in green coffee poured with hot water was tested, the caffeine value for Robusta was 0.186–0.293 g/L, while for Arabica: 0.006–0.188 g/L. The caffeine content of 0.006 g/L was obtained in decaffeinated coffee. The brewing method for each coffee sample was the same, but the coffees differed in both the origin, the time of grinding, and degree of grinding. Robusta had a higher concentration of caffeine—the maximum average value was 1.3 times higher than the maximum concentration in Arabica coffee. 

Tfouni et al. (2014) [71] showed in a Robusta brew about 1.4–1.7 times higher than in Arabica coffee (light and medium roasted, respectively), prepared with the same method (pouring water and bringing to a boil). Meanwhile, in the study by Merecz et al. (2018) [65], the highest caffeine content in Arabica coffee, obtained by pouring hot water over coffee grounds, was similar to the concentration of caffeine in Robusta coffee, respectively: 0.700 ± 0.050 g/L (Arabica) and 0.760 ± 0.060 g/L (Robusta). When the coffee percolator method was used, similar values were found by the same authors: 0.650 ± 0.050 g/L (maximum caffeine content in Arabica) and 0.690 ± 0.030 g/L (caffeine content in Robusta). The difference may result from the origin and variety of the coffee. 

Fărcaş et al. (2014) [69] and Dankowska et al. (2017) [66] also obtained higher results for Robusta when the hot water pouring method was used. There were: 2.581 g/L for Robusta and 1.876 g/L for Arabica (about 1.4 times greater caffeine content) in the former study [69] and 0.602 ± 0.069 for Robusta, 0.375 ± 0.021 g/L for Arabica (1.6 times greater value) in the latter one [66]. The grains used by Dankowska et al. (2017) differed in origin [66].

On the other hand, in the case of coffee prepared in a coffee percolator by Dankowska et al. (2017) [66], the content of caffeine in the Robusta brew was approximately 1.8 times higher (0.892 ± 0.079 g/L—Robusta, 0.506 ± 0.036 g/L—Arabica).

As can be seen, almost all researchers confirm that Robusta contains about 1.4 to 1.8 times more caffeine.

### 2.2. The Impact of Brewing Time

Brewing time largely depends on the method of brewing [67,83]. This is due to the original sensory qualities of coffee brews that can be obtained after a certain period of time [84,85]. Time is not a decisive factor in influencing caffeine content, which is explained below.

In this research, the shortest brewing time is characteristic for coffee made in an espresso machine: 3 times for 13 s or 0.42 min. On the other hand, the longest brewing process concerned coffee prepared using the cold brew method: 282 or 420 min. Among all Arabica coffees brewed in a coffee machine, the brew prepared by Ludwig et al. (2014) [70] had the highest concentration: 7.908 g/L (0.174 g/22 mL), whereas the lowest value for cold brew coffee was recorded in the study by Rao et al. (2020) [61]: 1.036 ± 0.019 g/L. Considering only brewing time as a variable, it can be concluded that coffee made in a coffee machine, despite shorter brewing time, contained significantly more caffeine than cold brew coffee. However, these brews differ from each other in terms of other parameters and factors used.

Analysis of studies on Arabica coffee brewed in an espresso machine reveals that the highest caffeine content in dark roasted ground coffee: 7.908 g/L (0.174 g/22 mL) was reported by Ludwig et al. (2014) [70], and the lowest in roasted coffee 0.330 ± 0.020 g/L, as described in the study by Merecz et al. (2018) [65]. The effect of time was not taken into account in either paper. The amount of coffee used by Ludwig et al. (2014) [70] was: 20.4 g of ground coffee and the cup volume was 22 mL, while in the study by Merecz et al. (2018) [65] it was 2 g of coffee and 100 mL of water. The volume per cup is not specified. It can be assumed that the time of brewing in the paper by Ludwig et al. (2014) [70] was about 25 s, as in other similar research projects. However, in the study by Merecz et al. (2018) [65], the brewing time could have been longer, due to the larger amount of water used.

Differences in caffeine content may be caused by the amount of ground coffee and water, but brewing time does not appear to have a significant effect on its content. In the above-mentioned study by Ludwig et al. (2014) [70], the amount of water used was not specified, but it can be assumed that it was similar to the amount of brew obtained, while in the study by Merecz et al. (2018), prolonged extraction of espresso coffee was probably necessary because the amount of water was relatively large. That resulted in a dilution of the brew, which, given the small amount of ground coffee used, contributed to a much lower concentration of caffeine.

An espresso from Robusta coffee was prepared by only two of the research teams, we have analyzed: Ludwig et al. (2012) [72] and Merecz et al. (2018) [65]. The values obtained were: 2.533 ± 0.020 g/L [72] and 0.150 ± 0.010 g/L [65]. The caffeine concentration in the Robusta brew in the former study was lower than the above-discussed values for Arabica coffee but higher than for the Arabica brew in the same study by this author. In the study by Merecz et al. (2018) [65], the reason for low caffeine concentration could be, as mentioned, the small amount of coffee in relation to the amount of water used. The caffeine content recorded by Ludwig et al. (2012) [72] may have resulted from the larger volume of the brew compared to the Arabica brew in Ludwig et al. (2014) [70] (espresso regular extraction). The volumes were: 47 [72] and 22 mL [70], respectively.

Caprioli et al. (2015) [67], Ludwig et al. (2012) [72], and Ludwig et al. (2014) [70] also tested the effect of extending the extraction time of espresso coffee on the content of specific substances in the brew. Caprioli et al. (2015) [67] showed that with the extension of extraction time, the content of some compounds in espresso coffee, including caffeine, decreased. In the first four time periods, i.e., up to the volume of 25 mL, 85.46% of the total caffeine content was extracted for Arabica and Robusta and 84.31% for Arabica. Extending the extraction time to 40 s did not increase caffeine concentration, but diluted the coffee. The present research confirms that the adopted volume of 25 mL for traditional espresso coffee is favorable for the extraction of caffeine [67,70,72]. When assessing the content obtained during extraction, Ludwig et al. (2012) [72] found that the concentration of caffeine amounted to: 0.297 ± 0.002 and 0.040 ± 0.001 mg/100 mL in the first 0–8 s and 16–24 s, respectively [72]. In another study by Ludwig et al. (2014) [70], extending espresso extraction time, and thus increasing the volume of the brew itself from 22–23 mL to 43–55 mL, also resulted in increased caffeine extraction and its increased content in the brew: from 0.152–0.174 g/serving (22–23 mL) to 0.202–0.232 g/serving (43–55 mL) [70].

In the case of a filter coffee machine, Arabica and Robusta in the study by Ludwig et al. (2012) [72] and Arabica in the study by Caporaso et al. (2014) [68] had lower values than brews prepared in an espresso machine, despite longer brewing time (2–6 min) [64,67,68,70,72]. The exception is the study by Merecz et al. (2018) [65], in which the concentration of caffeine in espresso coffee was lower (from 0.330 ± 0.020 to 0.410 ± 0.020 g/L). The values obtained by Ludwig et al. (2012) [72] for Arabica coffee were as follows: 1.414 ± 0.024 g/L (coffee machine, 7 g of coffee, brew volume: 47 mL, brewing time: 3 times 0.13 min, no information on temperature or water) and 0.571 ± 0.001 g/L (filter coffee machine, 36 g of coffee, brew volume: 532 mL, brewing time: 6 min, water temperature: 90 °C). For Robusta, the amounts and brewing time used were similar to those applied to prepare Arabica coffee in the same study [72]. The values obtained were: 2.533 ± 0.020 g/L for coffee prepared in an espresso machine and 1.153 ± 0.004 g/L for a brew from a filter coffee machine.

Caporaso et al. (2014) [68] obtained higher caffeine content in filter coffee machine brews compared to the study by Ludwig et al. (2012) [72]: 1.390 ± 0.300 g/L (25 g of coffee, 300 mL of distilled water, water temperature: 90 °C, the volume of brew: 230 mL, brewing time: 2 min). The brewing time reported by Caporaso et al. (2014) [68] was shorter compared with that used by Ludwig et al. (2012) [72]. Thus, it seems that factors other than brewing time were important. 

Taking into account the coffee percolator method, the longest coffee brewing time was 15 min (Dankowska et al. (2017) [66]), and the shortest 2.13 min (Angeloni et al. (2018) [64]). The highest proportion of caffeine of all the coffees brewed in a coffee percolator expressed as g/L was obtained for an Arabica brew in a paper by Caporaso et al. (2014) [68]: 1.680 ± 0.200 g/L (brewing time: 3 min, 11.3 g of coffee, 80 mL of distilled water, the volume of brew: 62 mL, water temperature: 100 °C). The concentration of the examined substance in the Dankowska study (2017) [66], where the authors used the longest brewing time, was not the highest. The lowest result for this method of brewing was obtained by Merecz et al. (2018) [65]: 0.340 g/L (2 g of coffee, 100 mL of water, cold water brought to the boil, time unknown). Unfortunately, Merecz et al. (2018) [65] did not record the preparation time, although since cold water was used, it can be assumed that it took longer than when using hot water. Angeloni et al. (2018) [64] noted lower concentration compared to that obtained by Caporaso et al. (2014) [68]: 1.280 ± 0.040 g/L. The brewing time was slightly shorter in Angeloni et al. (2018) [64], however, there were also differences in the amount of coffee and water used, as well as in the origin of the coffee itself. In the case of Robusta coffee brew in Dankowska et al. (2017) [66], the amount of caffeine was: 0.892 ± 0.079 g/L. In Merecz et al. (2018) [65], where a Robusta brew was also prepared, caffeine content was slightly lower: 0.690 ± 0.030 g/L, which may be due to a different type of brewing. It can be concluded that the longer brewing time in Dankowska et al. (2017) [66] did not increase the content of caffeine. It seems that brewing time for coffee made in a percolator was also not a significant factor that contributed to the differences in caffeine content. 

As regards the method of pouring ground coffee by hot water, the authors of the studies under discussion obtained different values of caffeine. The longest brewing time was used by Dankowska et al. (2017) [66]—15 min, and the shortest by Fărcaş et al. (2014) [69]—2 min.

In the case of Arabica brews, the obtained caffeine concentrations were as follows: from 0.375 ± 0.021 g/L in Dankowska et al. study (2017) [66] to 1.876 g/L in the Fărcaş et al. study (2014) [69], for Robusta brews: from 0.602 ± 0.069 g/L [66] to 2.581 g/L [69]. Despite the longest brewing time, Dankowska et al. (2017) [66] obtained the lowest concentration, while Fărcaş et al. (2014) [69] the highest. As far as Arabica and Robusta blends are concerned, the highest value was also achieved by Fărcaş et al. (2014) [69]: 2.519 g/L, and the lowest by Ranić et al. (2015) [73]: 0.700 ± 0.110 g/L. The way in which Ranić et al. (2015) [73] prepared coffee differed from the other methods involving hot water described in the analyzed projects. The authors added coffee directly to hot water (95–100 °C) and boiled the brew again for a few seconds. This method is rather rare among consumers. Comparing coffee blends in terms of caffeine content will not be reliable due to the unknown proportions of Arabica and Robusta.

It can be concluded that brew time was not an important factor that influenced caffeine content in the brew. The reason for the differences could be the amount of ground coffee and water used, the origin and variety of the coffee, as well as the degree of grinding and roasting of the beans (not included in the above studies). Additionally, in the study by Fărcaş et al. (2014) [69], the original amount of coffee used was 1 g, while the authors recalculated the content as 5 g of coffee/100 mL of water to obtain the amounts drank by the consumer. This could have distorted the final results as depending on the amount of the substance, extraction in the brew varies.

Cold brew is becoming an increasingly popular method of brewing coffee. The brew is prepared at room temperature—from 20 to 25 °C or lower. The entire brewing process takes from several to even 24 h. This coffee has a characteristic taste and aroma due to the long-brewing time [61]. The caffeine content in brews made using this method was investigated by Rao et al. (2020) [61] and Angeloni et al. (2018) [64]. The former team obtained the following caffeine concentrations for coffee brewed for 6 h (20 g of coffee, 200 mL of distilled water, room temperature of water): from 1.036 ± 0.019 g/L for medium-roasted Arabica to 1.962 ± 0.041 g/L for dark roasted Arabica. Angeloni et al. (2018) [64] obtained a slightly higher value after 5.5–6 h (25 g of freshly ground coffee, undefined degree of roasting, 250 mL of water, water temperature: 20 °C): 1.250 ± 0.120 g/L. These values are lower than some of those yielded by the traditional hot water pouring method for Arabica. It could be evidence that other factors are important, while brewing time itself does not play a significant role.

Another method of preparing coffee is to brew it in a French press. This is slightly similar to pouring hot water over coffee, but the filter plunger prevents ground coffee from getting into the brew [86]. In the study by Rao et al. (2020) [61], Arabica coffee made in this way (20 g of coffee, 200 mL of distilled water, water temperature: 100 °C) had from 1.035 ± 0.039 to 1.095 ± 0.065 g/L of caffeine, depending on the degree of roasting. The preparation time for the brew was 6 min. Angeloni et al. (2018) [64] obtained significantly lower results for Arabica: 0.520 ± 0.060 g/L (15 g of coffee, 250 mL of mineral water, water temperature: 93 °C). Brewing time was reduced by a minute. It can be assumed that the difference in brewing time was not so great as to cause the caffeine content in both brews to differ as much as twofold, so most likely other factors (such as coffee variety, degree of grinding, amount of coffee, and water) affected the final caffeine content.

In the studies by Tfouni et al. (2014) [71] and Niseteo et al. (2012) [52], brewing time was not specified, therefore they have not been analyzed in this aspect.

In summary, high caffeine values can be achieved with a shorter brewing time, if other factors come into play, including temperature or pressure (e.g., in an espresso machine or coffee percolator) discussed later in this work. It follows that brewing time has an impact on caffeine content in a brew due to the longer or shorter contact of ground coffee with water, but is not an important factor. In addition, to reliably compare the effect of brewing time on caffeine content, the same variables (variety and type of coffee, amount of coffee, and water) should be used along with different brewing times.

### 2.3. The Impact of Temperature of Water

Water temperature can have a significant impact on caffeine content due to the fact that caffeine is moderately soluble in water at 20 °C (1.46 mg/mL). Moreover, caffeine’s solubility increases at 80 °C (to the value of 180 mg/mL), reaching its peak at 100 °C (670 mg/mL) [87]. It can be assumed that lower temperatures may slow down the extraction of caffeine in a brew.

Espresso is prepared in coffee machines that contain volumetric pumps, responsible for achieving appropriate temperatures (between 92–94 °C) and pressure (the most common being about 9 bar). When water flows through the filter with pressed coffee, many bioactive substances are extracted in the brew [88].

A study by Caprioli et al. (2014) [67] investigated the effect of temperature on caffeine extraction in espresso brewing. It was noticed that the increase in temperature from 88 °C to 92 °C during the brewing of a Robusta and Arabica blend led to an increase in the content of caffeine in the cup. On the other hand, at 98 °C, less caffeine was extracted, regardless of the level of pressure. In the case of Arabica, the total amount of caffeine also rose with increasing temperature (from 88 °C to 92 °C), regardless of the pressure. The authors concluded that the best conditions for caffeine extraction for espresso coffee were 92 °C at 7 and 9 bars.

Salamanca et al. (2017) [89] also showed that lowering the temperature (from 93 °C to 88 °C), when brewing coffee in a coffee machine, contributed to the reduction of caffeine extraction. On the other hand, a rise in temperature (from 88 °C to 93 °C) increases the amount of caffeine extracted in a brew, although Massella et al. (2015) [90] did not find any influence of temperature on caffeine content in brews in the case of the capsule method. The temperature used was 75–85 °C. It can be assumed that extraction would have been more efficient at higher temperatures. These studies were not included in our review as the species of coffee used was not given.

The results achieved by Rao et al. (2020) [61] and Angeloni et al. (2018) [64] allow us to infer that lower temperatures slow down the extraction of caffeine in a brew. The study used the same amounts of coffee and water, and the same types of coffee. After 7 h, Arabica coffee brewed at room temperature (about 20 °C), had similar levels of caffeine to coffee (with the same degree of roast) brewed at 100 °C for 6 min in the same study. However, in the case of the cold brew, using coffee ground at the highest roasting temperature, the result was significantly higher than for coffee brewed in the French press using hot water. The results obtained in this study were: 1.036 ± 0.019 to 1.962 ± 0.041 g/L (cold brew) and 1.035 ± 0.039 to 1.095 ± 0.065 g/L (French press). However, in the study by Angeloni et al. (2018) [64], the same variables were also used, except for the temperature and time of brewing. The authors noticed that coffee brews prepared in a French press for 5 min (water temperature: 93 °C) had lower caffeine content than cold brew coffee (water temperature 22 °C): 0.520 ± 0.060 g/L (French press) and 1.250 ± 0.120 g/L (cold brew), respectively.

In the analyzed publications, the temperature of water mixed with ground coffee varies from 90 °C to 100 °C. It was noticed when the temperature of water increased above 90 °C, the caffeine content in the brew also grew [52,62,63,65,66,71]. As mentioned earlier, caffeine solubility increases above 80 °C. To assess the differences in the effect of temperature, the same variables would have to be used with varying temperature levels. Owing to the fact that the solubility of caffeine raises with increasing temperature, this substance is extracted much faster when water at a temperature above 80 °C is used.

### 2.4. The Impact of Water Pressure

Pressure is one of the factors that can make a difference in caffeine content in brews obtained by the following brewing methods: coffee machine, coffee percolator, and (to a lesser extent) the Aeropress.

Caprioli et al. (2015) [67] analyzed the effect of different pressure values (7, 9, 11 bar) on caffeine content in Arabica and Arabica and Robusta blend espresso coffees (5% Arabica, 95% Robusta). The maximum results were obtained at a pressure of 7 bar and water temperature of 92 °C for an Arabica and Robusta blend: 10.303 g/L, while for Arabica at a pressure of 9 bar and water temperature of 92 °C: 5.270 g/L (0.132 g/25 mL). In the case of the Arabica and Robusta blend, the authors concluded that the increase in pressure at constant temperature resulted in slightly slower caffeine extraction, especially at 11 bar. In the case of Arabica, it was found that the increase in pressure may have had a minimal effect on caffeine extraction at a constant temperature. According to the researchers, the best conditions for the extraction of Arabica and Robusta blends are 92 °C at 7 bar, while for Arabica: 92 °C at 9 bar.

Comparing the results obtained by Caprioli et al. (2015) [67] with those reported by Ludwig et al. (2014) [70], where there was a constant pressure of 9 bar, Arabica coffee in the latter study [70] contained slightly more caffeine: 6.609–7.908 g/L. However, the amount of coffee used there was also higher: 7.5 g of coffee grounds in the study by Caprioli et al. (2015) [67] and 18.1–20.4 g of coffee in Ludwig et al. (2014) [70], which probably influenced the obtained caffeine values.

In the study by Angeloni et al. (2018) [64], a pressure of 9 bar was used. Compared to Ludwig et al. work (2014) [70], the obtained caffeine values were lower: 4.100 ± 0.160 g/L (0.122 ± 0.005 g/30 mL) for the classic espresso method and 4.200 ± 0.090 g/L (0.076 ± 0.002 g/18 mL) in an espresso specialty. One of the reasons may be the use of a smaller amount of coffee and a larger amount of brew in the classical method. However, in the espresso specialty, the amount of coffee used and the brew volume were more similar to those applied for the regular extraction by Ludwig et al. (2014) [70]. Angeloni et al. (2018) [64] used 18 g of coffee, brew volume: 18 mL. This may be due to other factors such as the degree of grinding ([70]: undefined, [64]: fine to coarse grinding) and roasting of coffee beans (Ref. [70]: light to dark roasted, Ref. [64]: undefined), and origin (Ref. [70]: Brazil, Ref. [64]: Ethiopia) [64,70]. It follows that pressure may influence the extraction of substances, including caffeine, but an increase in pressure—more than 9 bar—has not been shown to have a significant effect on increasing caffeine content in brews.

The above trend was confirmed in other publications: Parenti et al. (2014) [91], Masella et al. (2015) [90], and Andueza et al. (2002) [92]. In the study by Parenti et al. (2014) [91], new methods of preparing coffee in a machine were used: Hyper Espresso Method (HIP, capsules), I-Espresso System (capsules), and the conventional espresso machine method (CM, using ground coffee). The traditional method and HIP differed in terms of the pressure used: 9 bar and 12 bar respectively, but also as regards the amount of coffee: 14.5 ± 0.2 g (CM) and 6.7 ± 0.1 g (HIP). The authors did not provide information on the pressure in the I-Espresso System method; therefore it was not taken into account. The contents of caffeine in both brews were similar: 2.22 ± 0.55 (CM) and 2.31 ± 0.19 (HIP) mg/mL (0.002 g/L). The authors concluded that the methods used did not differ in their effect on the extraction of caffeine into the brew. However, it is worth noting that despite the lower coffee content in the capsule in the HIP method, caffeine concentration in the brew was similar to that obtained by the CM method, where twice the amount of coffee was used. The higher pressure may have intensified the extraction process of caffeine into the brew. However, the coffees used may have differed in terms of caffeine concentration in the beans, depending, for example, on variety.

The study by Masellaet al. (2015) [90] also showed no effect of pressure increase in the case of the capsule method on the content of caffeine in brews (pressure range 15–20 bar). The obtained caffeine values ranged from 2.16 ± 0.30 to 2.39 ± 0.26 mg/mL (about 0.002 g/L). Similarly, Andueza et al. (2002) [92] did not notice any effect of higher pressure on the extraction of caffeine in the brew (pressure range approximately 7 bar, 9 bar, 11 bar): 2.0 ± 0.03, 2.05 ± 0.03, and 2.01 ± 0.05 mg/mL (about 0.002 g/L). These studies were not included in this literature review due to the lack of information about species of coffee.

The pressure may influence the extraction of caffeine in brews prepared in a coffee percolator and an Aeropress. However, to draw conclusions, one would need to compare different brewing methods using the same variables. In the studies by Merecz et al. (2018) [65] and Dankowska et al. (2017) [66], the same amounts of coffee and water were used for all methods. Merecz et al. (2018) [65] found lower caffeine values (from 0.340 ± 0.020 to 0.650 ± 0.050 g/L) in Arabica coffee prepared in a coffee percolator, compared to Arabica coffee poured with hot water (from 0.410 ± 0.050 to 0.700 ± 0.050 g/L). However, compared with espresso coffee, depending on the coffee sample, brews prepared with the use of a percolator were characterized by lower, similar, or higher values (content for coffee machine: from 0.330 ± 0.020 to 0.410 ± 0.020 g/L). For Robusta coffee, a brew prepared in a coffee percolator also had less caffeine than coffee poured with hot water, respectively: 0.690 ± 0.030 g/L (coffee percolator) and 0.760 ± 0.060 g/L (pouring water), but more than a brew made in a coffee machine: 0.150 ± 0.010 g/L. The use of cold water for preparing a coffee percolator brew could have had an effect compared with the other methods where the water temperature was about 90 °C. The steam that escaped from the heated water for a long time could have made the coffee cake ‘clump’, thus making extraction difficult.

In the study by Dankowska et al. (2017) [66], coffee prepared with the use of a percolator contained more caffeine than coffee poured with hot water, correspondingly, for Arabica: 0.506 ± 0.036 g/L (coffee percolator) and 0.375 ± 0.021 g/L (pouring water), for Robusta: 0.892 ± 0.079 g/L (coffee percolator) and 0.602 ± 0.069 g/L (pouring water).

On the other hand, in the studies by Angeloni et al. (2018) [64] and Caporaso et al. (2014) [68], the authors used different amounts of coffee grounds and water for all coffee brews, therefore it is difficult to evaluate the influence of pressure. In the case of the coffee percolator, the pressure was determined at 1.5 bar by Angeloni et al. (2018) [64]. This brew had a slightly higher caffeine content than the cold brew coffee, the French press brew, and the Aeropress brew. Concentrations of caffeine achieved in this study were respectively: 1.280 ± 0.040 g/L—coffee percolator brew, 1.250 ± 0.120 g/L—cold brew, 0.520 ± 0.060 g/L—French press brew, and 0.780 ± 0.090 mg/L—Aeropress brew. Caporaso et al. (2014) [68], obtained a higher amount of caffeine in the coffee percolator brew than in those made in an American coffee maker (filter coffee machine) and Neapolitan pot brews, but lower than in coffee prepared in a coffee machine. Correspondingly: 1.680 ± 0.200 g/L coffee percolator brew, 1.390 ± 0.300 g/L American coffee maker brew, and 1.300 ± 0.180 g/L Neapolitan pot brew. Higher values in the coffee percolator may be due to the presence of pressure, but this is not clear as the amounts of coffee and water used were different.

However, there are no data to determine the potential effect of pressure on caffeine content in the Aeropress brewing method. An Aeropress consists of a cylinder, filter rod, and piston. Coffee brewing consists of creating pressure on the coffee cake through the piston, i.e., the upper part of the device [86]. In the study by Angeloni et al. (2018) [64], the pressure was specified as 1 bar, similar to the French press method. It follows that pressure does not play a significant role, but there is not enough research in this area to assess this.

### 2.5. The Impact of Roasting

Roasting coffee is a very important process that modifies the content of bioactive coffee compounds, affecting its sensory properties. What occurs is, among other things, degradation of polysaccharides, oligosaccharides, especially sucrose, as well as chlorogenic acids and trigonelline [42,93,94,95]. High temperature contributes to the formation of a number of volatile and non-volatile substances. The characteristic taste of roasted coffee results from non-enzymatic browning reactions, which include caramelization and the Maillard reaction [96,97]. The roasting process takes place at a temperature of 200 °C to 260 °C, depending on the degree of roasting described. There are 4 levels of roasting: light, medium, medium-dark and dark [98]. During the roasting process, green coffee beans almost double their volume [96,97], and their weight is reduced by about 15–25%, most of which is vaporized water [99].

Caffeine is an alkaloid that is thermally stable [100,101]. Some of it is lost during the roasting process, but a small part may be lost during the sublimation process [26,102]. In addition, changes in the microstructure of coffee beans occur during roasting. The pores close, which contributes to the accumulation of inorganic gases inside the beans. The pressure inside increases, which causes them to crack (characteristic crackling sounds), and, along with the roasting gas, a small amount of caffeine may also be released [103,104,105]. Caffeine losses may be greater at higher roasting temperatures [106].

The degree or temperature of roasting were determined only in the studies by Rao et al. (2020) [61], Tfouni et al. (2014) [71], Caporaso et al. (2014) [68], Ludwig et al. (2014) [70] and in one sample in the study by Merecz et al. (2018) [65].

In the study by Rao et al. (2020) [61], Arabica coffee was roasted at 194 °C, 203 °C, and 209 °C. Caffeine concentrations in cold brew coffee were similar in the case of coffee roasted at 194 and 203 °C (1.114 ± 0.056 g/L and 1.036 ± 0.019 g/L), while in the case of coffee roasted at 209 °C, caffeine content was higher (1.962 ± 0.041 g/L). On the other hand, the caffeine concentration in the traditional French press brew was similar, regardless of the roasting temperature: from 1.035 ± 0.039 to 1.095 ± 0.065 g/L. The authors concluded that the degree of roasting did not affect the caffeine content in the brew. The higher caffeine content in the cold brew method in the case of coffee roasted at 209 °C was not explained by the authors of the paper but could have been due to differences in the degree of grinding.

Ludwig et al. (2014) [70] also used beans: light, medium, and dark roasted. The authors did not notice significant differences in caffeine content or trends, respectively: 6.609 g/L—medium roasted coffee (211 °C), 7.174 g/L—lightly roasted coffee (197 °C), 7.908 g/L—dark roasted coffee (219 °C). They regarded the ratio of caffeine to chlorogenic acids as a good marker in determining the degree of roasting of coffee beans due to their greater thermal stability. Additionally, dark roasted coffee had a slightly higher caffeine content.

In the study by Tfouni et al. (2014) [71], Arabica and Robusta brews, despite the use of different roasting levels: light, medium, and dark roasted (roasting time: 7–12 min, 200 °C), also did not differ in terms of their caffeine content.

Macheiner et al. (2019) [63], measured caffeine content in green coffee brews. The obtained results ranged from 0.139 ± 0.002 to 0.188 ± 0.007 g/L and were lower than the concentration of caffeine in brews of roasted coffee prepared with the same method [62,65,66,69,71]. The differences may have arisen from the method of preparing the brew and the amounts of coffee and water used. The degree of grinding of the green coffee beans and their density related to the roasting process were also taken into account. 

Merecz et al. (2018) [65] and Caporaso et al. (2014) [68] used only medium roasted coffee beans. Therefore, the impact of the roasting process cannot be assessed.

There are also publications in which caffeine content decreased with the degree of roasting. In the study by Król et al. (2020) [45], the concentration of caffeine was the highest in lightly roasted coffee and decreased along with increasing roasting degree. Hečimović et al. (2011) [47] obtained similar results. Crozier et al. (2012) [102] also showed that the level of caffeine decreased in the coffee brew (pouring hot water) by about 80% during roasting. The decrease in caffeine is influenced by both types of roasting: at high temperature for a short time and at low temperature for a long time. On the other hand, Jeon et al. (2017) [107] did not notice any effect of the degree of roasting on caffeine content in coffee beans and brews. These three studies were not taken into account in this review due to the lack of information on specific species [45,106] and laboratory brewing methods [47].

### 2.6. The Impact of Grinding Degree

It seems that the time elapsed since the coffee beans were ground did not affect caffeine content, but it could influence the volatile matter. Freshly ground coffee contains more of it, which is why it is so desirable among consumers [108]. On the other hand, the degree of grinding of coffee beans plays an important role in the extraction of caffeine into a brew [109]. Moreover, the selection of the degree of grinding is largely related to the method of brewing [105,110,111]. There are 4 degrees of grain grinding: coarse, medium, fine, and very fine [83]. It is assumed that the longer the brewing time, i.e., the contact between water and coffee, the coarser the ground should be. For example, very finely ground coffee is used for Turkish coffee, which gives it a distinctive aroma and taste. On the other hand, for pressure and filter coffee machines, slightly less finely ground coffee beans are used due to the shorter brewing time [83,109,110]. In the case of coffee brewed in a French press, the beans should be more coarsely ground, depending on brewing time, which usually takes a few minutes [83].

Finely ground coffee has a smaller particle size and thus a larger contact surface with water. Depending on the method of brewing, this may have a positive or negative effect on the extraction of the substance. In the case of coarse coffee, the coffee cake (ground coffee) shows greater porosity and particle size, which in turn causes high porosity fraction, i.e., the flow of water through the ground coffee beans [110,111]. As a result, both the extraction and diffusion of the substance into the brew decrease—due to the small contact surface between large coffee particles and hot water [59,110,111,112,113,114]. The uniformity of grinding, i.e., the distribution of coffee particles of different sizes, is also important. This is because the extraction of substances from fine and larger coffee particles is different. Therefore, it affects the quality of the brew [112].

The authors of the studies analyzed in this review did not always take into account the degree of grinding. In the study by Macheiner et al. (2019) [63], green coffee came from various sources and had different degrees of grinding. The authors noticed that Arabica in coffee bags showed the lowest caffeine extraction efficiency of about 58%, which may be explained by the fact that it was the most coarsely ground. The researchers concluded that ground green coffee particles have a higher density and thus a smaller contact surface with water.

In the study by Tfouni et al. (2014) [71], ground coffee had a particle size of 400 µm or less, which seems appropriate for their methods: pouring water (25 °C), bringing it to a boil, and filtering through a paper filter, or pouring hot water 92–96 °C and using a paper filter. Angeloni et al. (2018) [64] also adapted the degree of grinding to the brewing methods used: fine grinding for classical espresso, espresso specialty method, and coffee percolator, and coarse grinding for cold brew, Aeropress, and French press method. Similarly, Caporaso et al. (2014) [68] used the same particle size for all brews: 350 µm for American coffee, Neapolitan pot, coffee machine, and coffee percolator. A higher degree of grinding is suitable for these methods. The appropriate degree of grinding certainly facilitated the extraction of substances, including caffeine.

Studies involving the degree of grinding and extraction of substances for brew mainly concern espresso coffee [59,75,115,116]. Derossi et al. (2018) [75] took into account 3 degrees of grinding in their study: fine, fine-coarse, and coarse. They demonstrated that caffeine content in brews (espresso, Turkish coffee, American coffee) was higher, the more coarsely the coffee was ground. These authors obtained the following caffeine concentrations (about 0.002 g/L): 2.47 mg/mL (fine), 2.68 mg/mL (fine-coarse) and 2.92 mg/mL (coarse) for espresso coffee, 2.01 mg/mL (fine), 2.10 mg/mL (fine-coarse) and 2.21 mg/mL (coarse) for Turkish coffee, 1.43 mg/mL (fine), 1.57 mg/mL (fine-coarse), 1.65 mg/mL (coarse) for American coffee. In turn, Andueza et al. (2003) [115] noted an inverse correlation for an Arabica and Robusta blend. The content of caffeine in espresso coffee increased with the degree of grinding, respectively (about 0.003 g/L): 3.05 mg/mL (coarse), 3.19 mg/mL (fine), 3.80 mg/mL (very fine). Bell et al. (1996) [116] also obtained the highest values of caffeine in brews (boiled coffee and filtered coffee) for fine ground coffee, respectively: 0.40 mg/mL (fine), 0.35 mg/mL (medium), 0.20 mg/mL (coarse). Similarly, Khamitova et al. (2020) [59] and Jeon et al. (2017) [106] found that the level of coffee grind influences the caffeine content in the cup. In the former of the two studies, caffeine concentration in espresso was higher when the particle size was 200–300 µm [59]. Jeon et al. (2017) [106] also noticed that the concentration of caffeine increased, respectively, from coarse to fine coffee powder in coffee prepared with the use of a dripper.

Consumers who prefer freshly ground coffee usually own home coffee grinders of varying power. A study by Murray et al. (2015) [117] aimed to show the time after which coffee is ground with a home grinder for the greatest amount of caffeine extraction in a brew. The authors noticed a positive correlation between the grinding time up to 42 s and the amount of caffeine in a coffee brew prepared in a filter coffee machine but did not notice such a correlation after a longer grinding time (84 s). This is important data when analyzing the consumption of caffeine by consumers. Research proves that the degree of grinding may affect the extraction of caffeine in the brew.

### 2.7. The Impact of Type of Water

In most of the analyzed studies, distilled water was used. The exceptions are, for example, the study by Angeloni et al. (2018) [64], where mineral water was used, and by Ranić et al. (2015) [73], where tap water was used. Both these types of water had an unknown mineral content. Moreover, Rao et al. (2020) [61] and Fărcaş et al. (2014) [69] used deionized water, while Macheiner et al. (2019) [63]: ultra-high quality water. 

Some studies did not take into account the type of water [62,65,67,68,70,71,72]. It seems that water has no effect on the extraction of caffeine itself, but may affect the quality and taste of coffee [118]. There is little research in the literature on this issue. Water can affect the taste and aroma of coffee due to its electrolyte content. It was noticed that distilled water, devoid of electrolytes, excessively emphasizes the acidity of coffee [119], while water rich in alkaline ions neutralized acidity [120]. Moreover, water rich in carbonates and bicarbonates with excessive content of sodium ions may extend brewing time [121]. As far as chlorination and hardness of tap water are concerned, it has been shown that these factors may, to some extent, change the taste of coffee and its quality by affecting the extraction temperature [122]. On the other hand, Navarini et al. (2010) [118] showed that the content of bicarbonate ions could affect the texture, volume, and durability of the froth in espresso coffee. More studies are needed to assess the effect of the type of water on the extraction of caffeine in a brew and the quality of the brew itself.

### 2.8. The Impact of Coffee/Water Ratio

The ratio of coffee powder to water seems to be an important factor. The authors of the studies presented in this paper used different amounts of coffee and water, which ultimately affected caffeine content in the brews they prepared. Some of the determinations do not include the amount of water used. In the case of espresso, it can be assumed that the amount of water used should be similar to the volume of the brew. The ratios of the amount of coffee to the water used in the case of espresso coffee for Arabica were as follows: 18.1–20.4 g/22–23 mL brew [70], 18.1–20.4 g/43–55 mL brew (over-extraction) [70], 14 g/30 mL brew (classical espresso), 18 g/18 mL brew (espresso specialty) [64], 7.5 g/25 mL brew [67], 7 g/25 mL brew [68], 7 g/46–47 mL brew [72], 2 g/100 mL water [65]. It can be seen that Ludwig et al. (2014) [70] used the largest amount of coffee powder in relation to the volume of brew and thus obtained the highest concentration of caffeine for Arabica coffee: 7.908 g/L (0.174 g/22 mL) among all coffees brewed in the coffee machine. Merecz et al. (2018) [65] used the smallest amount of coffee, only 2 g, and as much as 100 mL of water. The small amount of coffee powder and the large amount of water contributed to the low caffeine content: 0.330 ± 0.020 g/L (which equates to about 0.010 ± 0.001 g/25 mL of espresso). In the case of Robusta, the ratios of coffee powder to water were: 7 g of coffee/46 mL of brew [72], 2 g/100 mL of water [65], which gave the following amounts of caffeine: 2.533 ± 0.020 g/L [72] and 0.150 ± 0.010 g/L of brew [65].

As regards the coffee brewed in a filter coffee machine, the following amounts were used: 25 g/300 mL of water (230 mL volume of brew) in the study by Caporaso et al. (2014) [68] and 36 g/532 mL of brew (600 mL of water) in the study by Ludwig et al. (2012) [72]. However, it can be estimated that the amount of coffee used in relation to the amount of water in the study by Caporaso et al. (2014) [68] was greater than that used by Ludwig (2012) [72], and the obtained caffeine content was also more than twice as high: 1.390 ± 0.300 g/L (0.173 ± 0.037/125 mL) and 0.571 ± 0.001 g/L, respectively (after converting about 0.071 g/125 mL). It follows that the amount of coffee used may have played a role.

In brew made by pouring hot water over ground Arabica coffee, the ground coffee/water ratios were as follows: 50 g/500 mL [71], 4 g/100 mL [66], 3 g o/200 mL [63], 2.5 g/150 mL [62], 2 g/100 mL [65], 1 g/100 mL [69].

Despite the lowest coffee/water ratio, Fărcaş et al. (2014) [69] obtained the highest value of caffeine: 1.876 g/L, i.e., approximately 0.281 g/150 mL per standard cup of brew. However, the authors used 1 g of coffee and converted the obtained caffeine concentration to 5 g per 100 mL of brew, hence they probably overestimated caffeine content. In the study of Tfouni et al. (2014) [71], where the ratio of coffee powder to water was the highest, caffeine content was 1.110–1.225 g/L for pouring water, which gives about 0.167–0.184 g/150 mL of the brew and for paper filter coffee: 0.873–0.990 g/L, i.e., about 0.131–0.149 g/150 mL of brew. The lowest caffeine concentration was noted by Merecz et al. (2018) [65]: from 0.410 ± 0.050 to 0.700 ± 0.050 g/L, where the coffee powder/water ratio was one of the lowest, right after the Fărcaş et al. (2014) [69] and Dąbrowska-Molenda (2019) [62].

In the case of Robusta coffee, the amounts were as follows: 50 g/500 mL [71], 4 g/100 mL [66], 3 g/200 mL [63], 2 g/100 mL [65], and 1 g/100 mL [69]. As in the case of Arabica coffee, Fărcaş et al. (2014) [69] obtained the highest caffeine content: 2.581 g/L (0.387 g/150 mL of brew), but as mentioned earlier, this amount may not be reliable. In Tfouni et al. (2014) [71], the concentration of caffeine was from 1.713 ± 0.057 to 1.920 ± 0.141 g/L, i.e., about 0.257–0.288 g/150 mL of the brew. On the other hand, the lowest result was obtained by Dankowska et al. (2017) [66]: 0.602 ± 0.069 g/L (about 0.090 g/150 mL), although the lowest coffee grounds/water ratio, just after that reported by Fărcaş et al. (2014) [69], was noticed by Merecz et al. (2018) [65]. This may be due to factors such as the degree of grinding or the origin of the beans.

Among the researchers using Arabica and Robusta blends, Caprioli et al. (2015) [67] also obtained the highest caffeine content (10.303 g/L, per cup: 0.258 g/25 mL) for espresso coffee (ratio: 7.5 g/25 mL of brew). On the other hand, Ranić et al. (2015) [73], using 3.4 g of coffee/100 mL of water, obtained the following caffeine content: 0.700 ± 0.110 g/L. In this study, a slightly different method of brewing was also applied (adding coffee to hot water and boiling it), which could have had an impact on the result.

For cold brew coffee (Arabica), the ratio of coffee powder to water was 25 g coffee/250 mL water in Angeloni et al. (2018) [64] and 20 g coffee/200 mL water in Rao et al. (2020) [61]. The ratios of ground coffee to water were the same in both studies. As mentioned earlier, caffeine values obtained by the researchers were similar, except for coffee roasted at 209 °C in the study by Rao et al. (2020) [61].

When comparing Arabica coffee brews prepared in a French press with hot water, the following amounts were used: 6 g/200 mL of water [61] and 15 g/250 mL of water [64]. Rao et al. (2020) [61] detected twice as high a concentration of caffeine. Thus, other factors could have been important, such as the degree of grinding (specified only in Angeloni et al. (2018) [64] as coarse), the degree of roast (not specified in the Angeloni et al. (2018) [64]), and the origin of the coffee (Ethiopia—[64], Colombia—[61]) or the variety of Arabica coffee (not specified in either study).

Different amounts of coffee and water were also used to make Arabica coffee brews, prepared in a percolator: 11.3 g of coffee/80 mL of water [68], 15 g/150 mL of water [64], 4 g/100 mL [66], 2 g/100 mL [65]. In the study by Caporaso et al. (2014) [68], the largest amount of coffee was used in relation to the amount of water, which is related to a caffeine concentration of, respectively: 1.680 ± 0.200 g/L (0.067 ± 0.008 g/40 mL). The lowest coffee powder/water ratio was used by Merecz et al. (2018) [65], resulting in the lowest concentration of caffeine: from 0.340 ± 0.020 to 0.650 ± 0.050 g/L, which after conversion to 40 mL is, respectively, 0.014–0.026 g/40 mL of brew. It follows that in Merecz et al. (2018) [65], apart from the difference in the method of brewing (cold water heated to boil), the amount of coffee used (lower than in the other studies) also had an impact on the lower caffeine content.

Comparing the amounts used to make Robusta brew, in a percolator, Dankowska et al. (2017) [66] detected a higher concentration of caffeine (about 0.036 g/40 mL of brew) than Merecz et al. (2018) [65] (about 0.028 g/40 mL of brew), which is consistent with the use of more coffee, respectively: 2 g/100 mL—Merecz et al. (2018) [65] and 4 g/100 mL—Dankowska et al. (2017) [66].

The other methods used by Angeloni et al. (2018) [64] and Caporaso et al. (2014) [68] Aeropress and Neapolitan sweat differed in the amount of coffee and water used: 16.5 g of coffee/250 mL of water and 15.4 g/145 mL, respectively. Arabica coffee prepared in a Neapolitan pot had a higher caffeine content and, at the same time, a higher ground coffee/water ratio, 1.300 ± 0.180 g/L (0.052 ± 0.007 g/40 mL) for the Neapolitan pot and 0.780 ± 0.090 (0.093 ± 0.010/120 mL) for the Aeropress. However, these are two different brewing methods.

### 2.9. The Impact of Volume

Another important factor is the volume of the brew, which is drunk by the consumer. The brewing methods vary and therefore yield different volumes of beverage. Some authors did not provide information on caffeine content per cup. However, it seems to be significant from the practical point of view of a consumer who prepares coffee brews. When discussing caffeine content per cup, to compare the same brewing methods and different brewing methods, the amounts reported by researchers per g/L or otherwise were converted to method-specific volume.

In the case of espresso coffee, an Arabica (5%) and Robusta (95%) blend had the highest caffeine content per serving in the study by Caprioli et al. (2015) [67]: 0.258 g/25 mL (10.303 g/L). Robusta’s percentage was much higher was significant. As Arabica and Robusta blends are often used for brewing coffee in machines, more caffeine may be delivered to consumers. In this review, only one type of espresso using an Arabica and Robusta blend was considered.

On the other hand, in the study by Ludwig et al. (2014) [70], the Arabica espresso regular extraction obtained the highest content of caffeine per liter of brew, respectively: 7.908 g/L and 0.174 g/22 mL, while the highest content per serving of espresso over-extraction contained 0.232 g/43 mL (4.218 g/L). Some consumers prefer a prolonged espresso, called ‘espresso lungo’. It has a larger cup volume: from 100 to 250 mL [53]. The espresso brew described by Ludwig et al. (2014) [70] cannot be regarded as ‘lungo’ (because of a smaller volume), but it can be concluded that people choosing such a brew may consume more caffeine than those who drink a standard portion of espresso (25 mL).

Taking into account the classical espresso (CE) and espresso specialty (ES) in the study by Angeloni et al. (2018) [64], the latter type of espresso per liter contains slightly more caffeine per liter (4.200 ± 0.090 g/L—ES and 4.100 ± 0.160 g/L—CE). Due to the small volume of the brew (18 mL), the consumption of caffeine will be lower, respectively: 0.076 ± 0.002 g/18 mL (ES) and 0.122 g ± 0.005 g/30 mL (EC). Converting the results obtained by Merecz et al. (2018) [65], the concentration of caffeine in a single Arabica espresso was small: about 0.008–0.010 g of caffeine per 25 mL of espresso. This is probably the result of using only 2 g of coffee per 100 mL of water. Instead, for Robusta coffee, the values ranged, after calculation, from 0.004 g/25 mL in the study by Merecz et al. (2018) [65] to 0.063 g/25 mL in the study by Ludwig et al. (2012) [72].

The portion of the brew for the filter coffee machine was set at 125 mL, for calculation, as suggested by Angeloni et al. (2018) [64]. The concentration of caffeine ranged from 0.071 g in the study by Ludwig et al. (2012) [72] to 0.173 g in the study by Caporaso et al. (2014) [68]. A Robusta coffee brew in the study by Ludwig et al. (2012) [72] contained 0.144 g/125 mL of caffeine, less than the Arabica coffee brew.

In the case of coffee poured with water, the volume of the brew may differ, depending on the individual preferences. In this calculation, 150 mL was used as the volume of a standard cup, as in the study by Dankowska et al. (2017) [66]. When water was poured over coffee, the caffeine content for Arabica coffee ranged from 0.056 g/150 mL (0.375 ± 0.021 g/L) in the study by Dankowska et al. (2017) [66] to 0.281 g/150 mL in the study by Fărcaş et al. (2014) [69]. However, as mentioned earlier, this is most likely the result of the fact that the authors converted the results. High caffeine values were also reported by Tfouni (2014) [71]: 0.166–0.184 g/150 mL. Caffeine concentration in Robusta coffee ranged from 0.114 g in the study by Merecz et al. (2018) [65] to 0.387 g in the Fărcaş et al. (2014) [69] per 150 mL brew. A high content of caffeine was also obtained by Tfouni et al. (2014) [71]: 0.257–0.288 g/150 mL of brew.

Green coffee in the study by Macheiner et al. (2019) [63] yielded the lowest caffeine concentrations of all coffees prepared by pouring hot water, from 0.139 ± 0.002 to 0.188 ± 0.007 g/L of Arabica brew and 0.186 ± 0.008 to 0.293 ± 0.014 g/L of Robusta brew.

The caffeine content in cold brew coffee was converted by Angeloni et al. (2018) [64] to 120 mL of brew and amounted to 0.150/120 mL, respectively, while in Rao et al. (2020) [61], also after conversion to 120 mL of brew, correspondingly to Angeloni et al. (2018) [64], caffeine concentration was: 0.124–0.235 g (depending on the roasting temperature).

As regards the coffee percolator, a brew volume of 40 mL was taken, as in the study by Caporaso et al. (2014) [68] and Angeloni et al. (2018) [64]. In the case of Arabica coffee, the caffeine content per 40 mL of brew ranged from 0.014 in Merecz (2018) [65] to 0.067 g/40 mL in Caporaso et al. (2014) [68]. The amounts adopted by Caporaso et al. (2014) [68] seem to be more reliable with regard to consumer consumption. However, for Robusta, the values were 0.028 g in the study by Merecz et al. (2018) [65] and 0.036 g/40 mL in the study by Dankowska et al. (2017) [66].

In the case of French press coffee, the volume of brew was assumed to be 120 mL, as in the study by Angeloni et al. (2018) [64]. Caffeine concentrations in Arabica coffee ranged from 0.062 g in Angeloni et al. (2018) [64] to 0.131 g in Rao et al. (2020) [61].

For Aeropress coffee, the cup volume was set at 120 mL [64] and for Neapolitan pot at 40 mL [68]. Caffeine content was 0.093/120 mL and 0.052/40 mL, respectively.

To sum up, it follows that the most caffeine was found in the brews of Robusta coffee poured with cold water and boiled in the study by Tfouni et al. (2014) [71]: 0.288 g/150 mL of brew, followed by an espresso Robusta and Arabica blend: 0.258 g/25 mL in Caprioli et al. (2015) [67]. High caffeine values were also noticed in cold brew coffee, respectively: 0.150 g/120 mL [64] and 0.124–0.235 g/120 mL of brew [61]. The result obtained by Fărcaş et al. (2014) [69] was found to be less reliable due to the conversions used. In the case of espresso, the pressure on the coffee cake during the preparation process, called tamping, can also play an important role. The pressure affects the porosity of the coffee cake, and thus the extraction of the substance in the brew [111,123]. On the other hand, Kuhn et al. (2017) [113] did not observe any impact of tamping on the extraction of caffeine.

### 2.10. The Impact of Other Factors

Among other factors that may affect the content of coffee bioactive substances in the brew, the following can be distinguished: the influence of processing methods, coffee storage, as well as the origin of coffee, and the influence of environmental factors such as height above sea level and access to light.

Regarding the influence of geographical origin, it can also influence the caffeine content of the beans. Arabica coffee that grows in Kenya and Ethiopia has been shown to have a lower caffeine content than the same type of coffee that grows in Brazil. In the case of Robusta, a relationship was also noticed—the same species from Vietnam contained less caffeine than those from Uganda [44]. Environmental factors, e.g., light and height above sea level, also affect the content of substances in coffee beans, as demonstrated by Cheng et al. (2016) [38]. Light exposure is essential for the synthesis of caffeine inside the beans. The light demand of a coffee tree is not high, but it also depends on the species [38]. Some studies show that Robusta coffee growing in the dark is characterized by a lower level of caffeine in the beans, while in Arabica, limited exposure to sunlight may increase its content [124,125,126]. A study by Ribeiro et al. (2016) [127] also demonstrated that the caffeine content in the beans was slightly higher in the shade, but it was not a statistically significant value. Additionally, Somporn et al. (2011) [128] observed that coffee growing in the shade is characterized by a larger size and weight of beans, as well as a higher antioxidant activity related to the content of phenols and a higher content of chlorogenic acid [128]. This may be because growing in the shade means smaller changes in temperature, lower wind speed, and higher air humidity. The amount of sunlight that reaches the plant must be neither too small nor too great to ensure the right conditions for growth [129].

Altitude above sea level may also positively correlate with caffeine content [130]. Ribeiro et al. (2016) [127] noticed that Arabica coffee growing at an altitude of ≥1200 m above sea level had higher caffeine content than beans grown <1000 m above sea level, respectively: 13.39 to 12.35 g/kg of beans.

The way coffee is grown may also affect caffeine content, whether it is conventional or organic, as shown by Król et al. [45]. Conventional coffee has more caffeine than organic coffee; the authors found that freshly ground organic coffee contained 4.61 ± 1.69 mg/g of caffeine, while conventional coffee: 5.26 ± 1.97 mg/g. Nitrogen fertilizers are often used in conventional cultivation, while no artificial fertilizers or pesticides are used in the production of organic coffee [131,132]. It has been shown that nitrogen fertilizer, especially easily soluble ones, can increase the content of caffeine in coffee beans [132]. In the case of organic coffee, caffeine itself, which acts as a ‘natural pesticide’, helps fight pests [131].

Post-harvest processing like wet or dry processing and storage of coffee beans can also affect caffeine content. Coffee fruits are treated to remove the pericarp and then raw coffee beans are dried. There are two processing methods: wet and dry. In the wet method, ripe coffee fruits are mechanically cleaned of the pericarp. The residues are fermented and then washed off. In the dry method, coffee fruits are sun-dried and mechanically cleaned. After these processes, beans are dried and hulled to remove the endocarp, called parchment. Thus, the composition of coffee beans may vary, depending on the method used [133,134]. However, in the study of Joet et al. (2010) [135], wet processing did not have a statistically significant effect on caffeine content. Ribeiro et al. (2016) [127] also did not show a significant effect of wet or dry processing. Król et al. (2020) [45] investigated the effect of 12-month storage of roasted beans at 5 °C on caffeine content. They showed that the concentration of caffeine increased slightly in conventional coffee: from 5.26 to 5.41 mg/g, while a significant increase was observed in organic coffee from 4.61 to 8.55 mg/g. Detection of caffeine may be caused by the degradation of compounds—theaflavins and caffeine during storage [43]. Additionally, to obtain a product with very good sensory properties, varieties from the same species are often mixed before or after the roasting process. As a result, the individual varieties of coffee available in the stores may differ in terms of their content of individual bioactive ingredients, including caffeine [83]. The role of the discussed factors is summarized in Table 4.

The limitations of this literature review are as follows: the comparison of the research results may be biased due to the different ways the results are expressed by the authors (standardization was used to unify the results), different origins of coffee, and the use of two different methods for determining caffeine content by authors.

## 3. Conclusions

Coffee brews differ in terms of caffeine content. This can be influenced by various factors: the brewing method used, including brewing time, amount of coffee and water, type of water, cup volume, brewing temperature, pressure (mainly in the case of a coffee machine and coffee percolator), as well as the type and variety of coffee, its origin, and the degree of roasting or grinding of the beans. A specific tendency was noticed that coffee brewed in an espresso machine in particular studies had a higher amount of caffeine, which could be associated with the used coffee/water ratio. Espresso obtained the highest caffeine content per liter of brew, both in the case of Arabica, Robusta, as well as Arabica and Robusta blends. In most comparisons, green coffee contained less caffeine than roasted coffee prepared with the same method. The authors used different amounts of coffee and water, which, among other things, influenced the final results. Some researchers did not reveal caffeine concentrations per portion of the brew. It is important from the point of view of the consumer and the practical application of the obtained results. It seems that to effectively compare the influence of certain factors on caffeine content, the other variables should be kept constant. Moreover, it seems that many consumers drink coffee because of its taste and aroma. Therefore, the type of coffee and the method of brewing should be chosen according to preferences.

## Figures and Tables

**Table 1 foods-10-01208-t001:** Caffeine content (g/L) in 100% Arabica coffee brews.

Caffeine Content Av ± SD (g/L)	Methods	Time (min)	Amount of Coffee (g)	Amount of Water (mL)	Type of Water	Volume of Coffee Brew (mL)	Pressure (bars)	Temperature (°C)	Degree/Conditions of Roasting	Type of Coffee	Country	Methods of Analysis	References (Year)
1.962 ± 0.041	French press (cold brew)	420	20	200	DIw	Nd	Nd	Room t.	209 °C	G	Colombia	HPLC	[61] (2020)
1.114 ± 0.056									194 °C				
1.036 ± 0.019									203 °C				
1.095 ± 0.065	French press	6						100	194 °C				
1.056 ± 0.047									203 °C				
1.035 ± 0.039									209 °C				
0.489	Pouring water	5	2.5	150	Nd	Nd	Nd	100	R	FG	Nd	HPLC	[62] (2019)
0.188 ± 0.007	Pouring water	5	3	200	UHQw	Nd	Nd	100	Green	FG	Nicaragua	HPLC	[63] (2019)
0.183 ± 0.003											Bali		
0.175 ± 0.003											Guatemala		
0.173 ± 0.007											Mexico		
0.171 ± 0.001										G	Honduras		
0.167 ± 0.001										FG	Ethiopia		
0.166 ± 0.004										G	Brazil		
0.151 ± 0.010										FG	Tanzania		
0.139 ± 0.002										G (tea bag)	Nicaragua		
0.006 ^#^										FG	Honduras		
4.200 ± 0.090	Coffee machine—espresso specialty method (portafilter, La Marzocco GS3, Italy)	0.44	18	Nd	Mw	18	9	93	R	FG (fine course)	Ethiopia	HPLC-DAD	[64] (2018)
4.100 ± 0.160	Coffee machine—espresso classical method (portafilter. La Marzocco GS3, Italy)	0.45	14	Nd		30	9	93					
1.280 ± 0.040	Coffee percolator	2.13	15	150		40	1.5	100					
1.250 ± 0.120	Cold-brew	282	25	250		120	1	20		FG (coarse)			
0.780 ± 0.090	Aeropress	1.35	16.5	250		120	1	93					
0.520 ± 0.060	French Press	5	15	250		120	1	93					
0.410 ± 0.020	Coffee machine (portafilter, De’Longhi, EC145, Italy)	Nd	2	100	Nd	Nd	Nd	Nd	R	G	Brazil, Colombia, Central America	SP	[65] (2018)
0.390 ± 0.010		Nd							M	G	South/Central America, Brazil		
0.330 ± 0.020		Nd							R	G	Nd		
0.700 ± 0.050	Pouring water	10						90	M	G	South/Central America, Brazil		
0.470 ± 0.050		10							R	G	Nd		
0.410 ± 0.050		10							R	G	Brazil, Colombia, Central America		
0.650 ± 0.050	Coffee percolator	Nd						Cold water and heated to the boil	R	G	Brazil, Colombia, Central America		
0.420 ± 0.040		Nd							R	G	Nd		
0.340 ± 0.020		Nd							M	G	South/Central America, Brazil		
0.506 ± 0.036	Coffee percolator (brews were filtered)	15	4	100	Dw	Nd	Nd	100	R	FG	Costa Rica, Tanzania, Peru, Mexico, Guatemala	SP	[66] (2017)
0.375 ± 0.021	Pouring water (brews were filtered)												
5.270	Coffee machine (portafilter, Aurelia Competizione)	0.42	7.5	Nd	Nd	25	9	92	Nd	FG	Colombia	HPLC-VWD	[67] (2014)
5.231							11						
4.750							7						
4.512							7	98					
4.348							9						
4.172							11						
3.910							7	88					
3.851							9						
3.540							11						
2.440 ± 0.240	Coffee machine—espresso (fully automatic, Spinel Pinocchio C, Italy)	0.42	7	Nd	Nd	25	9.5	93	M	G	Italy	SPME-GC/MS	[68] (2014)
1.680 ± 0.200	Coffee percolator—moka	3	11.3	80	Dw	62	Nd	100					
1.390 ± 0.300	American coffee maker (filter coffee machine)	2	25	300	Dw	230	Nd	90					
1.300 ± 0.180	Neapolitan pot	5	15.4	145	Dw	75	Nd	90					
1.876	Pouring water	2	1 g (calculation for 5 g)	100	DIw	Nd	Nd	Hot water	R	G	Nd	SP	[69] (2014)
7.908	Coffee machine—regular extraction	Nd	20.4	Nd	Nd	22	9	92	D (219 °C)	G	Brazil	HPLC	[70] (2014)
7.174			18.6			23			L (197 °C)				
6.609			18.1			23			M (211 °C)				
4.489	Coffee machine—over-extraction		18.1			45			M (211 °C)				
4.218			20.4			55			D (219 °C)				
3.691			18.6			43			L (197 °C)				
1.225	Pouring water (25 °C), bringing to a boil and filtering through a paper filter	Nd	50	500	Nd	Nd	Nd	25 °C and coming to a boil	L	G	Brazil	HPLC	[71] (2014)
1.110									D				
1.108									M				
0.990	Paper filter							92–96	D (12 min, 200 °C)				
0.925									L (7 min, 200 °C)				
0.873									M (10 min, 200 °C)				
1.414 ± 0.024	Coffee machine (portafilter, Saeco Aroma, Italy)	3 * 0.13 *	7.0	45	Nd	47	Nd	Nd	Nd	FG	Guatemala	HPLC	[72] (2012)
0.571 ± 0.001	Filter coffee machine	6	36	600	Nd	532	Nd	90					
about 1.200	Coffee machine	Nd	7	Nd	Dw	50	Nd	95–97	R	G (capsules)	Nd	HPLC	[52] (2012)

* Three espresso fractions were collected sequentially every 8 s; D—dark roasted coffee, DIw—deionized water, Dw—distilled water, FG—freshly ground coffee, G—ground coffee, HPLC—high-performance liquid chromatography, HPLC-DAD—high-performance liquid chromatography with diode array detector, L—lightly roasted coffee, M—medium roasted coffee, Mw—mineral water, Nd—no data available, R—roasted coffee, SP—spectrophotometric method, UHQw—ultra-pure water, # decaffeinated coffee.

**Table 2 foods-10-01208-t002:** Caffeine content (g/L) in 100% Robusta coffee brews.

Caffeine Content Av. ± SD (g/L)	Methods	Time (min)	Amount of Coffee (g)	Amount of Water (mL)	Type of Water	Volume of Coffee Brew (mL)	Pressure (bars)	Temperature (°C)	Degree/Conditions of Roasting	Type of Coffee	Country	Methods of Analysis	References (Year)
0.293 ± 0.014	Pouring water	5	3	200	UHQw	Nd	Nd	100	Green	FG	India	HPLC	[63] (2019)
0.227 ± 0.010										G			
0.186 ± 0.008										G (tea bag)			
0.760 ± 0.060	Pouring water	10	2	100	Nd	Nd	Nd	90	R	G	Nd	SP	[65] (2018)
0.690 ± 0.030	Coffee percolator	Nd						Cold water and heated to the boil					
0.150 ± 0.010	Coffee machine (portafilter, De’Longhi, EC145, Italy)	Nd						Nd					
0.892 ± 0.079	Coffee percolator	15	4	100	Dw	Nd	100	100	R	FG	Indonesia, Yemen, India, and Vietnam	SP	[66] (2017)
0.602 ± 0.069	Pouring water							Nd	Nd	FG			
2.581	Pouring water	2	1 g (calculation for 5 g)	100	Dw	Nd	Nd	Hot water	R	G	Nd	SP	[69] (2014)
1.920 ± 0.141	Pouring water (25 °C), bringing to a boil and filtering	Nd	50	500	Nd	Nd	Nd	25 °C and coming to a boil	M	G	Brazil	HPLC	[71] (2014)
1.763 ± 0.061									D				
1.713 ± 0.057									L				
1.655 ± 0.049	Paper filter							92–96	M				
1.290 ± 0.225									L				
1.233 ± 0.278									D				
2.533 ± 0.020	Coffee machine (portafilter, Saeco Aroma, Italy)	3 * 0.13	7	45	Nd	46	Nd	Nd	Nd	FG	Vietnam	HPLC	[72] (2012)
1.153 ± 0.004	Filter coffee machine	6	36	600	Nd	532	Nd	90	Nd	FG			

D—dark roasted coffee, Dw—distilled water, FG—freshly ground coffee, G—ground coffee, HPLC—high-performance liquid chromatography, L—lightly roasted coffee, M—medium roasted coffee, Nd—no data available, R—roasted coffee, SP—spectrophotometric method, UHQw—ultra-pure water, 3 * 0.13—three times for 0.13 min.

**Table 3 foods-10-01208-t003:** Caffeine content (g/L) in a blend of Arabica and Robusta coffee brews.

Caffeine Content Av ± SD (g/L)	Methods	Time (min)	Amount of Coffee (g)	Amount of Water (mL)	Type of Water	Volume of Coffee Brew (mL)	Pressure (bars)	Temperature (°C)	Species	Degree/Conditions of Roasting	Type of Coffee	Country	Methods of Analysis	References (Year)
10.303	Coffee machine (portafilter, Aurelia Competizione)	0.4	7.5	Nd	Nd	25	7	92	Robusta blend (95% Robusta + 5% Arabica)	Nd	FG	Nd	HPLC-VWD	[67] (2014)
10.206							9	92						
9.171							11	88						
8.504							7	88						
8.052							11	92						
8.038							9	88						
6.432							7	98						
6.376							9	98						
4.448							11	98						
1.180 ± 0.100	Coffee added to hot water, boiling	3	6.9	100	Tap water	Nd	Nd	95–100	Arabica and Robusta blend	R	G	Brazil, India, Vietnam, African	SP	[73] (2015)
0.700 ± 0.110			3.4											
2.519	Pouring water	2	1 g (calculation for 5 g)	100	Dw	Nd	Nd	Nd	Arabica and Robusta blend	R	G	Nd	SP	[69] (2014)

Dw—distilled water, FG—freshly ground coffee, G—ground coffee, HPLC—high-performance liquid chromatography, Nd—no data available, R—roasted coffee, SP—spectrophotometric method.

**Table 4 foods-10-01208-t004:** Factors influencing caffeine content in coffee brews.

Factors	Possible Impact on Caffeine Content
Species	Robusta coffee has genetically more caffeine than Arabica
Brewing time	Not a decisive factor
Temperature of water	Caffeine is most soluble at 100 °C. A lower temperature reduces caffeine extraction
Water pressure	Not a decisive factor. Higher water pressure does not increase caffeine extraction
Roasting beans	Possible increase in caffeine loss during roasting, but the evidence is inconclusive
Grinding degree	The evidence is not conclusive, whereas the degree of grinding is closely related to the brewing method. It affects the aroma and taste of coffee, which is probably more important from the point of view of the consumer
Type of water	Probably does not affect caffeine extraction, but may affect the flavor and aroma of coffee
Coffee/water ratio	Probably has the greatest influence on caffeine content in the brew
Volume of coffee drink	Different brewing methods have a different volume, which affects caffeine content in the brew
Origin of coffee beans	The origin is related to climatic and environmental factors that may have an influence
Light exposure	The shade can have a positive effect on caffeine content in the coffee beans, but it is probably species dependent
Height above sea level	Possible positive effect on caffeine in Arabica beans. No data available on Robusta
Method of growing	The use of nitrogen fertilizers can increase the amount of caffeine
Storage of coffee beans	Not-significant influence of caffeine beans processing methods

## Data Availability

The analyzed publications are available from the authors.

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
