# Peer review of "Influence of Various Factors on Caffeine Content in Coffee Brews"

_foods, 2021, doi:10.3390/foods10061208_

Round 1
Reviewer 1 Report
The authors provide an impressive review on the caffeine content in coffee infusions dependent on various parameters. The main conclusion is that coffee extraction is a multifactorial process, but caffeine as excellent water soluble compound is mostly influenced by coffee-water ratio.
The following revisions are necessary:
The English must be corrected by a native speaker. There are many instances of Grammar mistakes etc
Abstract: perhaps also specify the “highest content” per portion
Line 19: perhaps give more details on espresso machine (portafilter type?)
Line 19: 7.5 g per what volume?
Line 27: The correct scientific term for robusta would be canephora, not the other way around. Robusta is a variety of canephora. Most commercial coffee sold as robusta is not even Coffea canephora var. robusta, but of other varieties (probably mostly hybrids).
Line 31-34: please re-check the coffee production data. This looks very low. Wikipedia reports that Brazil alone is producing more than 2 million metric tons per year. Is there a miscalculation from bags to tons?
Line 91: please specify the second “coffee machine” type. Better use “portafilter” throughout, and perhaps fully automatic machines if this is meant here?
Line 124 and throughout: please round all values off to full number and do not report commas. For example, considering measurement uncertainty, it makes no sense at all to state a number like 10,302.8 mg/L. Most preferably this might be reported as 10.3 g/L. Actually, this value might be re-checked. Is there a confusion between mg/kg in the powder and mg/L in the infusion?
Line 153: I disagree that canephora is characterized by a bitter taste. Do you mean caffeine itself? Please also try a pure specialty canephora. They taste extremely good. (I agree that most commercial material is of low quality with high number of defects).
Line 168: Implausible findings should be discussed in more detail. How can it be explained that arabica is higher than canephora? Was this already the case before extraction? Perhaps there was a mismatch in samples?
Line 242 and 244 and throughout: please check correct spelling of name Caprioli/Capriolli?
Line 908: I would not necessarily agree with the importance of pressure. A main conclusion could be that caffeine extraction is mostly dependent on coffee-water ratio as it is quite easily soluble in water. It could also be mentioned that caffeine is not everything. I actually would avoid over-extraction (e.g. to gain the maximum caffeine) for taste reasons.
Reviewer 2 Report
The manuscript entitled “Influence of various factors on the caffeine content in coffee infusions”, authored by Ewa Olechno, Anna Puścion-Jakubik, Małgorzata Elżbieta Zujko, and Katarzyna Socha, deals with the investigation of the main factors affecting the concentration of caffeine in coffee drinks. Since coffee drinks are the most consumed drinks in the world and widely distributed, the topic is really interesting and current. Furthermore, the manuscript is written with authority, and is very exhaustive. I have only small tricks:
Keywords should be words not contained in the title, at most present in the abstract. Their usefulness is to make easier the searching of the article using the common scientific search engines. Since several keywords are already present in the title, and/or repeated several times in the abstract, I strongly advise the authors to change some of the suggested keywords with other news. Authors can provide up to 10 different keywords.
Data reported in both main text and tables should be reported using the scientific system. In particular, authors should use the ‘dot’ and not the ‘comma’ as decimal separator (i.e. 1,962 -> 1.962). Moreover, the same number of significant figure should be reported for both means and standard deviations (i.e. 1.962 ± 41 -> 1.9 ± 41, or 1.96 ± 41.0). Please, fix it.
Section 2 should be removed. This section could be justified if the authors had performed meta-analyses. Since the article is a review, the authors do not need to justify the choice of their bibliographic references. Please, remove it.
Please change the title of section 3 to "factors affecting the caffeine content in coffee drinks", or something like that.
The subsections describing the various factors affecting the caffeine content are really well described and exhaustive. However, I find it difficult to make a point of the topic without a schematic representation. Authors should consider adding a diagram or scheme in which the different factors are reported, and how the caffeine content can vary according to these factors. This diagram could be placed at the beginning of section 3.
Round 2
Reviewer 1 Report
All my comments were adequately considered. However, please re-check the botanical name of "robusta". It should be Coffea canephora Pierre ex A. Froehner. I also still believe that most commercial material labelled as "Robusta" is NOT Coffea canephora var. robusta, but some other variety of canephora. Therefore, the scientific proper term throughout would be using canephora instead of robusta. This probably can be changed on page proof stage.
Reviewer 2 Report
After the revision, the manuscript can be now considered for the publication in Foods
